# Optimising Aripiprazole Long-Acting Injectable: A Comparative Study of One- and Two-Injection Start Regimens in Schizophrenia with and Without Substance Use Disorders and Relationship to Early Serum Levels

**DOI:** 10.3390/ijms26031394

**Published:** 2025-02-06

**Authors:** Giada Trovini, Ginevra Lombardozzi, Georgios D. Kotzalidis, Luana Lionetto, Felicia Russo, Angela Sabatino, Elio Serra, Simone Castorina, Giorgia Civita, Sara Frezza, Donatella De Bernardini, Giuseppe Costanzi, Marika Alborghetti, Maurizio Simmaco, Ferdinando Nicoletti, Sergio De Filippis

**Affiliations:** 1Clinica Villa von Siebenthal Neuropsychiatric Hospital, Via della Madonnina 1, 00040 Rome, Italy; giada.trovini@gmail.com (G.T.); simone.castorina@outlook.it (S.C.); giorgia.civita@hotmail.it (G.C.); frezza.sara9@gmail.com (S.F.); marika.alborghetti@uniroma1.it (M.A.); sergio.defilippis@me.com (S.D.F.); 2Department of Molecular and Developmental Medicine, Division of Psychiatry, School of Medicine, University of Siena, 53100 Siena, Italy; 3Department of Psychiatry, Fondazione Policlinico Universitario Agostino Gemelli IRCCS, LargoAgostino Gemelli 8, 00168 Rome, Italy; 4Department of Neuroscience, Section of Psychiatry, Università Cattolica del Sacro Cuore, Largo Francesco Vito 1, 00168 Rome, Italy; 5Clinical Biochemistry, Mass Spectrometry Section, Sant’Andrea University Hospital, Via di Grottarossa 1035-1039, 00189 Rome, Italy; luanalionetto@gmail.com (L.L.); donatelladb@yahoo.it (D.D.B.); giuseppecostanzi@gmail.com (G.C.); maurizio.simmaco@uniroma1.it (M.S.); 6SPDC DSM ASL BA (SPDC Ospedale Santa Maria Degli Angeli), P.za Padre Pio 21, 70017 Putignano, Italy; liciarusso9570@gmail.com; 7DSM ASL Viterbo Distretto C (CSM Civita Castellana), via Francesco Petrarca snc, 01033 Civita Castellana, Italy; angela.sabatino@asl.vt.it; 8Azienda Sanitaria Locale Lecce, UOC CSM Nardò, Via XXV Luglio, 4, 73048 Nardò, Italy; dottelioserra@yahoo.it; 9Department of Mental, Neurological, Dental and Sensory Organ Wellbeing, Fondazione PTV—Policlinico Tor Vergata, Tor Vergata University Hospital, Viale Oxford, 81, 00133 Rome, Italy; 10Department of Neuroscience, Mental Health and Sensory Organs (NESMOS), Faculty of Medicine and Psychology, University of Rome “La Sapienza”, Via di Grottarossa 1035-1039, 00189 Rome, Italy; 11Department of Physiology and Pharmacology “Vittorio Erspamer”, Faculty of Medicine and Pharmacy, University of Rome “La Sapienza”, Piazzale Aldo Moro 5, 00185 Rome, Italy; nicoletti@neuromed.it; 12IRCCS Neuromed, Pozzilli, Via Atinense, 18, 86077 Pozzilli, Italy

**Keywords:** aripiprazole long-acting injectable, aripiprazole serum levels, long-acting injectable antipsychotics, single injection start, two-injections start, schizophrenia, substance use disorders, medication adherence, therapeutic drug monitoring, clinical course

## Abstract

Aripiprazole as a long-acting injectable (LAI) is initiated in oral aripiprazole-stabilised patients and needs, after first injection, 14 days supplementation of oral aripiprazole (one-injection start, OIS). Recently, an alternative two-injection start (TIS) was advanced, involving two 400 mg injections with a single 20 mg oral supplementation of aripiprazole. We tested the two regimens in patients with schizophrenia (SCZ, *n* = 152, 90 men and 62 women) with (SUD^+^; *n* = 93) or without (SUD^–^; *n* = 59) substance use disorders (SUDs), comparing OIS (*n* = 66) with TIS (*n* = 86) and SUD^+^ vs. SUD^–^. For 26 patients, we measured weekly for one month, aripiprazole + dehydroaripiprazole (active moiety) levels. Patients were followed for three months after LAI with psychopathology and quality-of-life scales (BPRS, CGI-S, ACES, BIS-11, and WHOQOL). All groups improved in psychopathology with no differences between OSI and TIS and between SCZ–SUD^+^ and SCZ–SUD^–^. The TIS group was associated with serum blood levels of the active moiety within the therapeutic window, while the OIS group showed peaks above the window, possibly exposing patients to toxicity. Treatments were well-tolerated. Here we showed no disadvantages for TIS vs. OIS and possibly increased safety. Shifting the initiation of aripiprazole LAIs to the TIS modality may be safe and pharmacokinetically advantageous.

## 1. Introduction

Schizophrenia (SCZ) is a chronic psychiatric disorder, characterised by the persistence of impairment in cognitive and perceptual functions, behaviour, and affectivity. Approximately, its incidence is 1:10,000/year, its prevalence ≈1% worldwide, and its lifetime risk 1% [1]. SCZ is characterised by positive symptoms, such as hallucinations, delusions, and odd and bizarre behaviours, and negative symptoms, such as diminished emotional expression, flat affect, alogia, anhedonia, avolition, and asociality, as well as cognitive, emotional, and general symptoms that are often associated with decreased social functioning and general impairment [2].

The management of SCZ requires ongoing care for symptom control, relapse prevention, and psychosocial rehabilitation, and focuses primarily on antipsychotic treatment [3]. Antipsychotic medications generally work by blocking dopamine D_2_ receptors to control acute symptoms and to reduce the frequency and severity of relapses during maintenance [3]. First-generation antipsychotic drugs (FGAs) block dopamine D_2_ receptors by about 90% and can cause significant adverse effects, related to both ideation and movement (extrapyramidal side effects, EPSEs). Most prescribed antipsychotics are second-generation (SGAs) or atypical antipsychotics, blocking dopamine receptors more selectively than conventional antipsychotics, and decreasing the risk of EPSEs. Both FGAs and SGAs tend to be effective in reducing positive symptoms, whereas negative symptoms appear to be more resistant to antipsychotic action. EPSEs and dysmetabolic effects attributed to FGAs vs. SGAs are models not supported by clinical observation [4]; these adverse events occur with both generations of antipsychotics. Some drugs are not associated with EPSEs (e.g., clozapine), while others show fewer dysmetabolic effects (e.g., haloperidol and other butyrophenones, which may be related to cardiotoxicity leading to sudden death). This marketing-dictated terminology ignores chemical structure and has led to leakage from one category to another; for example, the substituted benzamide amisulpride, which is now claimed to belong to SGAs, fell among FGAs when it was synthesised [5]. The typical/atypical dichotomy had not been proposed at that time [6] and, when “atypicals” became fashionable, it claimed its place in the latter category [7]. The tendency to increase the number of antipsychotic drug generations continued even after the development of drugs with dopamine receptor modulatory activity. This class of drugs could partially activate dopamine D_2_ receptors in the prefrontal cortex and inhibit D_2_ receptors in the mesolimbic system. Some authors refer to these drugs as third-generation antipsychotics [8], but here we call them dopamine receptor partial agonists (DRPAs). DRPAs currently in clinical practice are aripiprazole (7-{4-[4-(2,3-dichlorophenyl)piperazin-1-yl]butoxy}-3,4-dihydroquinolin-2(1 H)-one), brexpiprazole (7-[4-[4-(1-benzothiophen-4-yl)piperazin-1-yl]butoxy]quinolin-2(1 H)-one), and cariprazine (N’-[trans-4-[2-[4-(2, 3-dichlorophenyl)-1-piperazinyl]ethyl]cyclohexyl]-*n*,*n*-dimethylurea). They all have a piperazinyl structure, which is also shared by the experimental drug TPN672 (7-(2-(4-(benzothiophen-4-yl)piperazin-1-yl)ethyl)quinolin-2(1 H)-one maleate) [9].

Long-term, continuous treatment with antipsychotic medications is important for achieving and maintaining symptom control [10]. Psychotic relapse is extremely distressing for patients and caregivers and is associated with multiple downstream effects on disease progression and brain structure, such as progressive grey and white matter structure and volume reduction [11]. Relapses can also lead to a reduced response to previously effective antipsychotics, potentially contributing to treatment resistance [12]. To overcome these issues, psychiatry has focused on “precision medicine” in recent decades. This concept considers individual variability to build evidence-based medicine that should entrain clinical practice [13]. Therapeutic drug monitoring (TDM) is a patient-centred management tool for precision medicine. According to drug properties and patient features, TDM enables the tailoring of drug dosing to the individual patient [14].

Long-acting injectable (LAI) antipsychotics were developed to overcome poor therapeutic compliance and the need for daily therapy. Due to their pharmacokinetic properties, LAIs should provide stable plasma/serum drug concentrations and accordingly reduce the risk of relapse. They have several advantages over oral antipsychotics, including the control of non-adherence, the reduction of pill burden, and the reduced consequences of treatment gaps [15]. Furthermore, LAIs may reduce patients’ self-stigma [16].

Considering that LAIs are effective and well tolerated in people with SCZ, improving adherence and reducing relapse, they deserve special consideration in people with comorbid SCZ and substance use disorder (SUD) [17]. SUDs are very common in young people with psychosis [18]. They may precede or follow the onset of psychotic symptoms [19]. Patients with SUDs often have worse premorbid conditions, earlier clinical onset, more severe psychotic symptoms, higher relapse and rehospitalisation rates, and more suicide attempts than patients without SUDs [20,21,22,23]. SCZ–SUDs comorbidity is associated with high rates of oral medication non-adherence [17].

Aripiprazole has been approved by the US Food and Drug Administration (FDA) in 2002 for the treatment of SCZ. It is a partial dopamine D_2_, serotonin 5-HT_1 A_ receptor agonist, and a serotonin 5-HT_2 A_ antagonist [24,25,26]. In particular, aripiprazole showed quite high affinity for dopamine D_2_, D_3_, and serotonin 5-HT_1 A_, 5-HT_2 A_, and 5-HT_7_ receptors (*K*_i_ values of 0.74–3.3 nM, 1–9.7 nM, 5.6 nM, 8.7–35 nM, and 10.3 ± 3.7 nM, respectively), a moderate affinity for serotonin 5-HT_2 C_, α_1_-adrenergic, and histamine H_1_ receptors (*K*_i_ values of 22–180 nM, 25–35 nM, and 25.1 ± 2.6 nM, respectively), moderate-to-weak affinity for the dopamine D_4_ receptor (*K*_i_ = 510 ± 93 nM), and low affinity for the serotonin transporter (*K*_i_ = 1080 ± 180 nM), while its affinity for muscarinic cholinergic and opioid receptors is negligible [26]. Furthermore, this drug increases the phosphorylation of glycogen synthase kinase 3β in various brain regions, differently from D_2_ receptor antagonists and other DRPAs [27,28].

As a DRPA and 5-HT_2 A_ receptor antagonist, aripiprazole exerts antipsychotic effects by decreasing dopaminergic neurotransmission in brain areas with hyperdopaminergic activity, such as the mesolimbic system; at the same time, it can enhance dopaminergic-mediated neurotransmission in brain areas with hypodopaminergic activity, such as the nigrostriatal and mesocortical systems. This mechanism may result in lower EPSE risk and in potential improvement of negative symptoms and cognitive deficits [29]. The effects of aripiprazole are thought to be due to both the parent drug and, to some extent, its major metabolite dehydroaripiprazole, which accounts for approximately 29–39% of the parent drug exposure in plasma. Aripiprazole is primarily metabolised hepatically via the cytochrome P450, particularly via isoenzymes CYP3 A4 and 2 D6 [30]. Dose adjustment may be indicated in individuals who are concomitantly exposed to drugs that induce or inhibit CYP3 A4 and/or 2 D6, or in those who carry allelic variants of the isoenzymes that make them poor, intermediate, fast, or ultra-rapid metabolisers [31]. Several oral formulations, including tablets and solutions, are approved by the FDA for the treatment of SCZ. An intramuscular injection form of aripiprazole is approved by the FDA and the European Medicines Agency (EMA) for the treatment of agitation associated with SCZ, and LAI forms are approved for the treatment of SCZ in adults [32].

We used the DRPA aripiprazole LAI initiation regimen of one 400 mg injection supplemented with 14 days of 10–30 mg oral aripiprazole treatment (OIS regimen) vs. an aripiprazole LAI initiation regimen of two 400 mg injections and 20 mg oral aripiprazole on day 1 only without the need for further oral supplementation (TIS regimen). Both regimens were followed by monthly 400 mg doses. Our aim was to evaluate the short-term differences in the efficacy and serum levels of the drug and its major active metabolites in patients with SCZ with or without comorbid SUDs (SCZ–SUD^+^ and SCZ–SUD^–^, respectively).

## 2. Results

### 2.1. Socio-Demographic Measures

We included a sample of 152 individuals (90 men and 62 women, with a mean age of 32.71 years, SD = 12.25). The mean age at onset was 20.51 years (SD = 7.26); 8.8% of patients were married, 79.6% were single, 6.8% were divorced, and 2.7% were widowed. The mean age at clinical onset was 20.67 years (SD = 7.29). Of the included patients, 66 (42 men and 24 women, with a mean age of 31.5 years, SD = 12.7), of which 43 had an SUD, received the aripiprazole LAI OIS regimen, while 86 (48 men and 38 women, mean age 33.6 years, SD = 11.9), 50 of which had an SUD, received the aripiprazole LAI TIS regimen. The demographic and clinical characteristics of the recruited patients are shown in Table 1. In our sample, there were significantly more single persons and significantly less married people than in the general population. Furthermore, patients who were assigned to the OIS regimen were significantly more impulsive and had significantly a better quality of life (QoL) on three of four World Health Organization Quality-of-Life (WHOQOL) dimensions than patients assigned to the TIS group (Table 1). Despite these baseline disparities, the OIS group was associated with significantly better QoL social relationships and less impulsiveness than TIS after one month, but the latter superiority of the OIS group, which was also present at baseline, was not present after three months of treatment, while QoL social relationships in OIS group, with respect to the TIS group, retained its advantage. However, after three months of treatment, the TIS group had a better response on the global severity and on the general psychopathology scale compared to the OIS group.

Of the 152 recruited patients, one male patient from the SCZ–SUD^–^ who was assigned to the TIS group discontinued treatment after 1 week due to onset of akathisia, while another five patients did not complete the follow-up. All these patients belonged to the SCZ–SUD^+^ group, three were men and two women, three were assigned to the OIS group and two to the TIS group. Drop-out patients decided to be followed up by their respective specific drug abuse services, thus withdrawing voluntarily from our study.

### 2.2. Comparison Between Aripiprazole LAI OIS vs. Aripiprazole LAI TIS Regimens in the Non-SUD Group

To investigate the effect of the different aripiprazole LAIs starting regimens on the clinical variables, we used a general linear model (GLM) with repeated measures, comparing the assessments before drug administration at BL, and at one and three months after injection. The analysis revealed significant main effects for all examined variables. Table 2 highlights the assessed clinical measures in the non-SUD group in both aripiprazole LAI regimens. Both LAI initiation regimens showed progressive decrease in scores on psychometric scales from baseline to endpoint, with no significant intergroup differences. On the social subscale of the WHOQOL, patients put on TIS scored higher than those put on OIS at the third month of treatment (Table 2).

### 2.3. Comparison Between Aripiprazole LAI OIS vs. Aripiprazole LAI TIS in the SCZ–SUD^+^ Group

Table 3 shows all clinical measures assessed in the SCZ–SUD^+^ group. TIS showed better overall effects than OIS.

OIS and TIS CGI-S scores at BL were 5.10 and 5.19, while at endpoint they were 3.95 and 2.96, respectively. Both groups showed significant decreases from BL to endpoint, but the TIS regimen showed a greater effect. Furthermore, patients with the OIS regimen scored better than patients on TIS on the WHOQOL psychological and social domains (Table 3). On the BIS-11 scale, TIS displayed a statistically significant reduction in the scores of attention and planning subscales and the total score than the OIS regimen. Similarly, the TIS group showed less craving on the VAScrav scale than the OIS group for both frequency and intensity in the SCZ–SUD^+^ group. Scores on the BPRS and ACES did not differ between the treatment groups.

### 2.4. Serum Concentrations of Aripiprazole, Its Metabolite and the Active Moiety After LAI Injections

Table 4 shows the serum concentrations of aripiprazole and its active metabolite dehydroaripiprazole, for the OIS and TIS groups. Table 5 shows the concentrations of the active moiety (aripiprazole plus its active metabolite). Both groups started with almost the same concentration at baseline (432.40 ng/mL for the OIS group and 431.20 ng/mL for the TIS group), due to the fact that the patients were already receiving the oral formulation of the drug. After the first week, the serum concentration in the OIS group tended to increase and reached a value of 534.23 ng/mL, which is above the therapeutic window. In the following weeks, drug concentrations tended to decrease, reaching 392.61 ng/mL at week four. By contrast, serum concentrations in the TIS group tended to decrease gradually, always remaining within the therapeutic window. At week four, the serum concentration of the active moiety in this group was 273.94 ng/mL.

The OIS group had significantly higher serum levels of both aripiprazole and dehydroaripiprazole than the TIS group at 2 weeks post injection (Table 4), while the active moiety was significantly higher in the OIS group than in the TIS group at both 1 and 2 weeks post injection (Table 5; Figure 1). Baseline serum aripiprazole concentrations did not differ between the OIS and TIS groups, but concentrations were supramaximal in the OIS regimen, whereas they were within the therapeutic window in the TIS group. Mean levels of the active moiety were supramaximal in OIS group only one week after the first injection (and in some patients at week 2), while serum aripiprazole levels were above the window at baseline and one and two weeks after the first LAI injection.

### 2.5. Adverse Events

During this study, no serious adverse event occurred, except the above-mentioned akathisia that led to discontinuation. All adverse events, such as nausea, headache, muscle pain, fatigue, and insomnia, were transient and mild and did not require specific treatment or discontinuation.

### 2.6. Drugs Prescribed at Discharge

The drugs prescribed to patients at discharge were similar in both the OIS and TIS groups (Table 6).

To conclude, we found OIS and TIS to overlap in their clinical effects for the first three months of LAI administration, but we found active aripiprazole serum levels with a TIS to stay within the therapeutic window, while the same levels were suprathreshold with an OIS.

## 3. Discussion

In this study, we found no differences between an OIS and a TIS in the non-SUD group with respect to symptoms assessed by the CGI-S, 24-item BPRS, WHOQOL, BIS-11, and ACES. We could conclude from our results that both regimens confirmed the efficacy of aripiprazole LAIs in symptom control for patients with acute exacerbations of SCZ by reducing scores on the different psychometric scales.

The alternative TIS regimen is now approved; however, there is little information on the effects and potential benefits of this regimen in clinical practice [33]. We aimed to evaluate the efficacy of the OIS vs. the TIS regimens in reducing psychotic symptoms and its ability to improve global clinical status. Comparing patients with and without comorbid SUDs allowed us to evaluate whether the aripiprazole LAI TIS regimen could be a good start option for patients with SCZ and SUD. Aripiprazole LAI is a long-acting antipsychotic that is initiated at a dose of 400 mg, followed by 14 consecutive days of oral aripiprazole administration to maintain therapeutic aripiprazole concentrations during the initiation of therapy [34]. Given that the medication is often prescribed to patients with no or partial adherence, the concomitant administration of oral aripiprazole for 14 days can be challenging and problematic. The initiation regimen of two injections plus a single 20 mg dose of oral aripiprazole on the first day is a new and simplified initiation strategy to reduce the need for 14 days of oral administration and to optimise the therapeutic benefits of aripiprazole LAIs in patients with SCZ [35]. Salvi et al. [36] reported an off-label use of aripiprazole LAI TIS in a 16-year-old adolescent with SCZ. The medication was optimally tolerated with unremarkable side effects. One month after treatment, global functioning and illness insight had improved, along with score reductions on the Positive and Negative Syndrome Scale (PANSS) [37] and the CGI.

A comorbid SUD is usually a barrier to treatment, as it reduces adherence to treatment for SCZ [38]. LAI antipsychotics may be an effective treatment option for SCZ–SUDs while providing a viable solution to improve adherence issues in this population [17]. Our data showed how aripiprazole LAI TIS regimen determined a significant improvement in clinical symptoms, particularly in the patient’s global status, impulsivity, and quality of life. With respect to craving assessed only in the SUD-comorbid population, self-rated craving scores decreased over time despite abrupt substance discontinuation and concomitant antipsychotic use.

In a recently published study [39], a population pharmacokinetic model was modified to simulate the serum concentration of aripiprazole following different initiation scenarios; the alternative aripiprazole LAI TIS initiation regimen proved to be best for maintaining active drug levels (aripiprazole plus dehydroaripiprazole) within their therapeutic 150–500 ng/mL window [40,41]. This study [39] showed similar pharmacokinetic profiles for the TIS and OIS regimens. Tolerability and safety assessment indicated that a TIS was not associated with more safety concerns than an OIS [35,41]. There was a high interindividual and intraindividual variability in aripiprazole/dehydroaripiprazole serum levels in our sample, but these did not apparently affect clinical response. This is consistent with what has already been reported for aripiprazole serum levels below 150 ng/mL, which maintains efficacy [40]. Our results showed that, in the TIS regimen, active moiety levels remain within the therapeutic window (Figure 1), while the OIS regimen showed an early peak that was higher than commonly recommended levels, potentially exposing the patient to overdosage and toxicity. No serious or persistent adverse events occurred during the study period, except for akathisia that occurred in the above-described patient and led to drop-out.

In spite of the need for adopting a convenient initiation regimen, reducing as much as possible the associated oral aripiprazole supplementation with the first 400 mg monthly LAI dose, there have been few attempts at addressing this need. The first proposition used a simulation model to study the pharmacokinetics of a TIS vs. an OIS [39] and found no safety issues with TIS in 2021. This was confirmed by a chart-review of 133 Italian patients receiving a TIS [35], which was published in 2023, after the EMA approval of TIS in Europe. Another chart review of 24 Italian patients reported similar results for safety [42]. Another study was conducted in Spain and involved a survey of 50 psychiatrists, who thought favourably about a TIS, but these were only impressions [43]. A Spanish study reported reduced healthcare service utilisation with a TIS [44]. Another three publications, case reports or series, reported favourable results, extending also to other psychotic diagnoses [36,45,46]. However, no study to date has investigated real-patient pharmacokinetics or the clinical outcome in patients with or without SUD. This is, to our knowledge, the first report of real patient pharmacokinetic data.

Aripiprazole LAI comes in two chemical formulations. The original is the Otsuka aripiprazole monohydrate, which was introduced in 2011 and first trialled in 2012 [47]; it obtained FDA approval in 2013 [48]. The alternative is the Alkermes aripiprazole lauroxil, a dodecanoate (a medium-chain fatty acid anion, the conjugate base of dodecanoic acid, also called lauric acid, thus recalling somehow the chemical name of the formulation), which was trialled between 2011 and 2014 [49] and obtained FDA approval in 2015 [48]. In the European Union, the EMA granted approval to aripiprazole monohydrate on 15 November 2013 [50], but it never approved lauroxil. However, the problem of the first 14 days of simultaneous administration of oral aripiprazole, that is needed with both formulations [51], due to the fact that aripiprazole needs about one week (median *t*_max_) to reach its maximum plasma concentrations (*C*_max_) [52], was not resolved until the introduction of alternative initiators, that were produced by Alkermes. It involved the production of a single dose (675 mg) nanocrystal formulation of aripiprazole lauroxil with smaller, faster dissolving particles, to boost initiation [53,54]. This formulation needs only one 30 mg oral administration and is not intended for repeated dosing; after the start, the usual monthly injections are used [55,56]. Thus, American psychiatrists have had the opportunity to start their aripiprazole LAI using only one oral 30 mg aripiprazole administration since the late 2000s/early 2010s, while in Europe this was not feasible. To overcome this hurdle, scientists at Otsuka and Lundbeck devised a method for population pharmacokinetic modelling and simulation in 2020 [57], which the authors published the next year [39]. This consisted in administering the regular aripiprazole monohydrate LAI 400 mg in two injections the same day along with one 20 mg oral aripiprazole (TIS). The two injections should involve different muscles, i.e., the deltoid and the gluteal muscles. The procedure is then followed by the regular monthly injections of 400 mg aripiprazole monohydrate. As with aripiprazole lauroxil, the goal of a TIS is to reach the desirable serum concentrations of the active moiety faster, thereby reducing safety or adherence issues, such as those that one could encounter with continuous oral coadministration. We have endorsed this practice here and compared it with the established OIS procedure regarding both clinical effect and pharmacokinetics. A TIS and aripiprazole lauroxil nanocrystal need to be compared for the same parameters, to obtain a clear picture of what occurs, but we feel that it is unlikely that the two pharmaceutical companies that produce the respective drugs will agree to conduct a blind trial. For the moment, we may rely on a TIS as a method to overcome the problems posed by concurrent oral aripiprazole administrations for 14 days. We also believe that the Otsuka/Lundbeck method has the advantage over the Alkermes that it does not need the production of a new formulation using advanced technology, which has its costs and is used just once, but allows for two injections of the same drug, thus reducing costs.

In conclusion, an aripiprazole LAI requires oral supplementation during the first 2 weeks of treatment, which may expose non-compliant patients to a higher risk of relapse. For this reason, the two-injection start regimen may be an interesting option to optimise therapeutic benefits in patients with SCZ and to reduce the duration of hospitalisation, allowing earlier discharge. However, to our knowledge, no other study has tested the efficacy of the TIS regimen in patients with SUDs in real-world conditions. Our study could provide some evidence of validity for a TIS that is still lacking, despite its authorisation [33]. The TIS has positively impressed the administering physicians [43]; whether it will supersede an OIS is still a matter of speculation. LAI aripiprazole formulations are currently diversifying, and their patient indications are widening [58]. We are currently unable to foresee tomorrow’s landscape for aripiprazole LAI formulations.

Our study has several limitations. Our sample size was small and needs to increase in order to draw valid conclusions. In addition, there were no comparison groups, such as patients with or without SUDs treated with antipsychotics other than aripiprazole or treated with a placebo. In this study, we investigated a single-site population, and this may limit the representativeness of the sample. In the SUD comorbid group, the used medications were often multiple, and there were not sufficient subsample sizes to allow us to analyse individual medications. The open, non-randomised design of our study adds to the limitations. Although the sizes of the TIS and OIS samples subjected to serum active moiety monitoring for one month (*n* = 13 each) were small for drawing conclusions, we believe that performing such monitoring is one of the strengths of this paper, as it is the first of its kind. Furthermore, another limitation emerged from the baseline disparity of QoL and impulsiveness between the OIS and TIS groups, despite random assignment. We cannot explain why this occurred.

## 4. Materials and Methods

We conducted an open-label study of inpatients and outpatients with a diagnosis of SCZ who underwent hospitalisation at Villa Von Siebenthal Neuropsychiatric Hospital (Genzano di Roma, Italy) between December 2021 and June 2022. We recruited adult patients (aged between 18 and 65 years) with a DSM-5/DSM-5-TR diagnosis of SCZ. All patients started as inpatients; they were discharged after 1 month and then followed as outpatients for the next 3 months.

Exclusion criteria were comorbid major psychiatric disorders other than SCZ; suicide risk (as assessed by the Columbia–Suicide Severity Risk Scale [C-SSRS]) [59], i.e., a score of 4 or 5 on the suicidal ideation items or any “yes” response to suicidal behaviour items); comorbidity with major organic diseases (autoimmune or systemic connective tissue disease, treatment-resistant hypertension, type 1 diabetes, metabolic syndrome, major cardiovascular disease, or major neurological disease); history of epilepsy, head injury, electroencephalographic (EEG) abnormalities, and neurodevelopmental disorders; intelligence quotient (IQ) < 75, as assessed by the Wechsler Adult Intelligence Scale (WAIS) [60], unwillingness to participate or inability to sign the informed consent by the patient or the legal guardian.

For all patients, socio-demographic and clinical data were available on a dedicated data sheet. SCZ and SUD (cannabis, synthetic cannabinoids, cocaine, amphetamines, opioids, ketamine/phencyclidine or other NMDA receptor inhibitors, khat and other cathinone alkaloids, and alcohol and polysubstance use disorder) were diagnosed using the Structured Clinical Interview for DSM-5—Clinician Version (SCID-5-CV) [61]. The psychopathology of 152 patients at baseline, 1 month, and 3 months was assessed using the Clinical Global Impressions–Severity Scale (CGI-S) [62], the 24-item Brief Psychiatric Rating Scale (BPRS) (original version) [63], (Italian version) [64], the World Health Organization Quality-of-Life Scale (WHOQOL) [65], and the Agitation–Calmness Evaluation Scale (ACES) [66]. Impulsiveness was assessed using the Barratt Impulsiveness Scale (BIS-11) [67]. Craving was assessed with the Visual Analog Scale for Craving (VAScrav) [68]. The latter rates craving from 0 (no craving) to 10 (the most intense craving in the patient’s experience). During the observation period, we used clinical face-to-face or telephone interviews for all recruited patients. Patients were regularly screened for drug use at baseline and throughout the study period.

After meeting inclusion criteria and not meeting exclusion criteria, patients were informed about the aims and methods of this study and gave their free and informed consent. This study was approved by the Ethics Committee of the Health Authority of Rome (study 331-306-00387; 19 November 2019). It was conducted in accordance with the Human Rights Principles adopted by the World Medical Association at the 18th WMA General Assembly, Helsinki, Finland, June 1964, and subsequently amended by the 64th WMA General Assembly, Fortaleza, Ceará, Brazil, October 2013.

### 4.1. Treatments

All medications used in this study followed clinical prescribing guidelines as recommended in each product’s monograph. Recruited patients were antipsychotic-naïve or -free (in washout from antipsychotics for at least 2 weeks) and immediately treated with aripiprazole after the recommended titration. Aripiprazole was used as oral 10–30 mg/day according to patients’ needs and as LAI at 400 mg/month (once or twice at the first administration, according to the group the patient belonged, and once monthly thereafter). If patients were taking other antipsychotic medications, they discontinued them and established treatment with oral aripiprazole after an appropriate washout period of at least 2 weeks prior to the start of their LAI. Once the appropriate dose for all patients was reached (based on clinical course and physician judgment), the regimen was maintained for at least 3 months. Patients did not take other antipsychotics or antidepressants during the entire study period but were allowed to take benzodiazepines or gabapentinoids for anxiety and insomnia and those with SUD medications specifically used for each SUD, i.e., methadone and naltrexone. Patients were assigned to the TIS and OIS regimens casually. The OIS patients took one 400 mg injection for the first month and 10 mg aripiprazole for 14 days, while the TIS patients received 800 mg consisting of two concurrent injections of 400 mg delivered at different sites, i.e., deltoid and gluteal muscles, for the first month plus a single 20 mg oral aripiprazole dose on the first day of the injections. Thereafter, all patients of both groups received the monthly 400 mg aripiprazole LAI treatment. The OIS or TIS regimen of aripiprazole administration was established by the caring physicians according to the individual patient’s psychopathological status.

Besides observing the response in the entire cohort of patient, we divided our population into two groups, based on substance use (SCZ–SUD^+^ and SCZ–SUD^–^).

### 4.2. Aripiprazole Serum Concentrations After LAI Injections

In a sub-cohort of patients (26 patients), serum aripiprazole and dehydroaripiprazole levels were measured at baseline (before the first drug injection), 1, 2, and 4 weeks after the first LAI injection. Since we have excluded any drugs that patients were taking other than the study drugs, we had no problems with drug–drug interactions which are known to exist when aripiprazole is co-administered with paroxetine, sertraline, venlafaxine, duloxetine, carbamazepine, and oxcarbazepine [41].

Blood samples were collected in 5 mL vials, immediately centrifuged to obtain serum samples, and stored at −80 °C within 5 min until analysed. Twenty microliters (μL) of serum sample were treated with 180 μL deproteinisation solution for 30 s and centrifuged at 14,000 rpm for 5 min. Two μL of the clean supernatant was injected into the chromatography system. Compounds were detected using an improved liquid chromatography-tandem mass spectrometry (LC-MS/MS) analysis method [69]. Briefly, HPLC analysis was performed using an Exion liquid chromatography system (Sciex, Foster City, CA, USA), which included a binary pump, autosampler, solvent degasser, column oven, and controller. The chromatographic separation was performed using a reversed-phase column (Kynetex^®^ 2.6 µm Biphenyl 100 Å pore size, LC Column 100 × 2.1 mm, Phenomenex, CA, USA) equipped with a safety guard precolumn (Phenomenex, Torrance, CA, USA) containing the same packing material. The mobile phase consisted of a solution of 0.1% aqueous formic acid (eluent A) and acetonitrile containing 0.1% formic acid (eluent B); elution was performed at a flow rate of 500 mL/min using a linear gradient from 0% to 100% eluent B in 4 min. The oven temperature was set at 50 °C. The injection volume was 2 µL and the total analysis time was 6 min. Mass spectrometry was performed on a 5500 QTrap system (Sciex, Foster City, CA, USA) equipped with a Turbo Ion Spray source. The detector was set to positive ion mode. The ion spray voltage was set to 5,000 V and the source temperature was 400 °C. The collision activation dissociation gas was set to medium, and nitrogen was used as the collision gas. The Q1 and Q3 quadrupoles were tuned for unit mass resolution. The instrument was set to multiple reaction monitoring mode. Data were acquired and processed using Analyst 1.7 software.

TDM is performed to ensure that blood concentrations of an administered drug are within a defined therapeutic window and do not reach toxic levels. TDM defines the therapeutic window. The therapeutic window is the range of plasma/serum drug concentrations that allow the drug to act optimally, by establishing a lower limit below which a drug-induced therapeutic response is relatively unlikely to occur and an upper limit above which tolerability decreases, or further therapeutic improvement is relatively unlikely to occur. The therapeutic reference range for aripiprazole is between 100 and 350 ng/mL [40], while for the active ingredient (aripiprazole plus dehydroaripiprazole), it is between 150 and 500 ng/mL, with a level of recommendation of 2 [levels of recommendation are 1 = strongly recommended, 2 = recommended (there is evidence of efficacy in the identified range), 3 = useful, and 4 = potentially useful] [41].

### 4.3. Statistical Analysis

We calculated frequency distributions and descriptive statistics to analyse the sample. We first performed preliminary analyses to verify the internal consistency of the instruments used in the study by calculating Cronbach’s alpha. We then performed descriptive analyses, which included calculating frequencies, means, and standard deviations of the personal information and variables studied, on the total sample and on the different samples separately.

Where possible, differences between the groups were tested using multivariate analysis of variance (MANCOVA), and the significance of the main effect was analysed using Pillai’s Trace index. Bonferroni’s correction of α for multiple comparisons was applied in the case of significance [70].

To evaluate the efficacy of aripiprazole LAI on the CGI-S, BPRS, WHOQOL, BIS-11, ACES, and VAS outcomes, we used a repeated measures general linear model (GLM) to evaluate changes over time. Because the main drug effect was significant for all dimensions studied, we separately performed repeated measures mixed linear models (MLMs) comparing the psychopathological course of patients initiated with the OIS regimen and those initiated with the TIS regimen.

Post hoc comparisons were performed using Bonferroni’s correction [70]. Nominal variables were analysed using the chi-squared test (*χ*^2^). SPSS v.27.0.1 for Mac software (IBM Co., Armonk, NY, USA, 2020) was used for all analyses. Significance was set at *p* < 0.05.

#### Serum Concentrations of Aripiprazole After LAI Injection

We performed a Student’s *t*-test to compare patients who received aripiprazole LAI 400 mg (13 patients) and those who received aripiprazole LAI 400 + 400 mg (13 patients). This was possible because the Shapiro–Wilk test [71] showed a normal distribution of the two samples (*W* = 0.9113; *p* = 0.09082 for OIS and *W* = 0.9061; *p* = 0.08593 for TIS).

Here also we performed post hoc comparisons using the Bonferroni correction. We used contingency tables for non-parametric variables, and the significance of the main effect was analysed by the *chi*-squared test. We used SPSS v.27 for Mac software to perform all statistical analyses (IBM Corporation, Armonk, New York, NY, USA, 2019). Significance was set at *p* < 0.05.

## 5. Conclusions

We found that aripiprazole LAI is a valid treatment option for SCZ, with or without SUDs. In particular, aripiprazole LAI was effective on psychotic symptoms, both positive and negative. Comorbid SUDs did not confer treatment-resistance in this study. The aripiprazole LAI TIS regimen was found to be suitable for patients with comorbid SUDs and psychotic disorders, as it did not increase craving for illicit substances after abrupt discontinuation (on the contrary, craving decreased during this study in the SUD group). This is the first study testing a TIS with real pharmacokinetics, not just simulations, and the first to compare TIS and OIS on clinical grounds. Should our data be confirmed by future studies, new clinical perspectives for the adoption of aripiprazole LAI TIS initiation regimen may emerge in the therapeutic horizon of SCZ comorbid with specific substance use disorders.

## Figures and Tables

**Figure 1 ijms-26-01394-f001:**
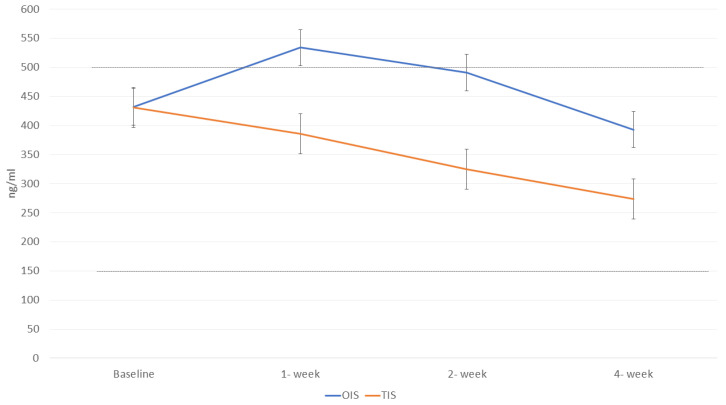
Course of aripiprazole plus dehydroaripiprazole serum levels during the first month post-injection in patients assigned to OIS (blue line) or TIS (orange line). OIS, one-injection start with aripiprazole LAI 400 mg plus oral supplementation of 10 mg/day aripiprazole for 14 days; TIS, two-injection start with two 400 mg aripiprazole injections plus one dose of 20 mg oral aripiprazole on the first day. The thick grey dashed lines represent the therapeutic window of the active moiety (between 150 and 500 ng/mL).

**Table 1 ijms-26-01394-t001:** Sociodemographic and clinical characteristics of the sample (*n* = 152).

	Total (*n* = 152)	OIS (*n* = 66)	TIS (*n* = 86)	*F*/*χ*^2^	*p*
Age, mean ± SD (years)	32.7 ± 12.2	31.6 ± 12.7	33.5 ± 11.9	0.451	0.637
Men/women	90/62	42/24	48/38	0.946	0.623
Age at onset, mean ± SD (years)	20.5 ± 7.3	20.6 ± 7.6	20.4 ± 7.0	0.014	0.986
Educational level (%)					
Illiterate	0.6%	Differential data not shown			
Primary school	36.8%				
High school	54.7%				
College/University	7.9%				
Marital status (%)					
Single	79.6%	Differential data not shown		49.476	**<0.00001 ***
Married/co-habiting	8.8%				
Divorced	6.8%				
Widowed	2.7%				
SCZ–SUD^–^/SCZ–SUD^+^	59/93	23/43	36/50	0.773	0.679
Baseline					
BPRS total, mean ± SD	66.1 ± 19.8	65.5 ± 17.0	66.5 ± 21.9	0.047	0.954
CGI-S, mean ± SD	5.2 ± 0.8	5.1 ± 0.7	5.3 ± 0.9	1.149	0.318
WHOQOL phys, mean ± SD	56.2 ± 12.1	60.1 ± 12.6	53.1 ± 10.8	6.505	**0.0017**
WHOQOL psyc, mean ± SD	47.6 ± 10.9	50.7 ± 11.7	45.2 ± 9.7	4.880	**0.0082**
WHOQOL soc, mean ± SD	48.9 ± 21.2	53.8 ± 16.9	45.2 ± 23.4	3.126	**0.0453**
WHOQOL env, mean ± SD	55.2 ± 12.3	56.8 ± 11.4	54.0 ± 12.5	0.989	0.373
BIS, mean ± SD	81.2 ± 13.8	84.4 ± 11.7	78.7 ± 14.7	3.259	**0.0398**
ACES, mean ± SD	2.3 ± 0.7	2.4 ± 0.6	2.2 ± 0.7	1.631	0.198
One month					
BPRS total, mean ± SD	44.8 ± 14.8	47.4 ± 13.6	42.7 ± 15.4	1.942	0.145
CGI-S, mean ± SD	3.6 ± 0.9	3.7 ± 0.7	3.6 ± 1.1	0.510	0.600
WHOQOL phys, mean ± SD	64.2 ± 11.6	66.1 ± 12.4	62.7 ± 10.7	1.550	0.214
WHOQOL psyc, mean ± SD	58.0 ± 12.2	60.5 ± 12.8	56.1 ± 11.4	2.449	0.08
WHOQOL soc, mean ± SD	55.5 ± 22.0	61.8 ± 17.8	50.6 ± 23.8	4.952	**0.007**
WHOQOL env, mean ± SD	63.8 ± 12.3	64.2 ± 13.2	63.5 ± 11.7	0.058	0.944
BIS, mean ± SD	68.8 ± 11.4	71.5 ± 11.7	66.6 ± 10.7	3.536	**0.030**
ACES, mean ± SD	3.2 ± 0.6	3.1 ± 0.6	3.3 ± 0.5	2.005	0.136
Three months					
BPRS total, mean ± SD	38.8 ± 12.8	42.2 ± 12.1	36.2 ± 12.8	4.121	**0.017**
CGI-S, mean ± SD	3.2 ± 1.0	3.5 ± 1.0	2.9 ± 1.0	5.506	**0.004**
WHOQOL phys, mean ± SD	63.2 ± 14.4	64.1 ± 15.1	62.5 ± 13.9	0.236	0.790
WHOQOL psyc, mean ± SD	58.8 ± 14.2	61.5 ± 14.3	56.8 ± 13.9	1.978	0.140
WHOQOL soc, mean ± SD	56.8 ± 23.0	63.9 ± 18.2	51.5 ± 24.8	5.409	**0.005**
WHOQOL env, mean ± SD	64.0 ± 14.5	64.4 ± 15.1	63.7 ± 14.2	0.036	0.964
BIS, mean ± SD	64.0 ± 13.5	65.2 ± 13.0	63.1 ± 13.8	0.446	0.640
ACES, mean ± SD	3.6 ± 0.6	3.6 ± 0.5	3.6 ± 0.6	0.020	0.980

Significant results in **bold**. * Compared to rates in the general US population, https://www.statista.com/statistics/242030/marital-status-of-the-us-population-by-sex/ (accessed on 19 December 2024). *Abbreviations:* ACES, Agitation–Calmness Evaluation Scales; BIS, Barratt Impulsiveness Scale; BPRS, 24-item Brief Psychiatric Rating Scale; CGI-S, Clinical Global Impressions–Severity; OIS, one-injection start long-acting aripiprazole regimen; SCZ, schizophrenia; SCZ–SUD^–^, patients with schizophrenia without a substance use disorder; SCZ–SUD ^+^, patients with schizophrenia with a substance use disorder; SD, standard deviation; SUD, substance use disorder; TIS, two-injection start long-acting aripiprazole regimen; WHOQOL, World Health Organization Quality-of-Life scale; -phys, physical; -psyc, psychological; -soc, social relationships; -env, environmental subscales.

**Table 2 ijms-26-01394-t002:** Repeated measures GLM in the non-SUD group (SCZ–SUD^–^
*n* = 59).

	Dose	BL	1 Month	3 Months	*F*	*p*
CGI-S	OIS	5.09	3.30	2.70	2.93	0.092
	TIS	5.47	3.67	3.08		
BPRS	OIS	69.39	48.17	38.57	0.33	0.566
	TIS	66.61	44.64	38.11		
WHOQOL (physical)	OIS	63.35	69.48	60.70	3.16	0.081
	TIS	53.72	62.06	62.72		
WHOQOL (psychological)	OIS	45.87	56.70	52.70	0.00	0.984
	TIS	44.06	55.19	56.17		
WHOQOL (social)	OIS	51.14	56.72	59.32	**4.17**	**<0.05**
	TIS	39.81	46.61	48.64		
WHOQOL (environmental)	OIS	57.74	60.96	59.57	0.02	0.896
	TIS	53.56	62.50	63.36		
BIS-11 (Attentional)	OIS	20.78	18.30	15.91	0.16	0.696
	TIS	21.00	18.11	16.97		
BIS-11 (Motor)	OIS	26.52	21.35	18.83	0.02	0.904
	TIS	25.86	21.25	20.06		
BIS-11 (Planning)	OIS	31.00	24.48	20.91	0.00	0.949
	TIS	28.75	24.61	23.25		
BIS-11 Total	OIS	78.52	64.35	55.57	0.01	0.938
	TIS	75.33	63.61	60.03		
ACES	OIS	2.57	3.13	3.83	0.24	0.625
	TIS	2.42	3.50	3.75		

Significant results in **bold**. *Abbreviations:* ACES, Agitation–Calmness Evaluation Scale; BIS-11, Barratt Impulsiveness Scale; BL, baseline; BPRS, 24-item Brief Psychiatric Rating Scale; CGI-S, Clinical Global Impressions–Severity scale; GLM, general linear model; OIS, one-injection start; TIS, two-injection start; WHOQOL, World Health Organization Quality-of-Life Scale.

**Table 3 ijms-26-01394-t003:** Repeated measures GLM in the SCZ–SUD^+^ group (*n* = 93).

	Dose	BL	1 Month	3 Months	*F*	*p*
CGI-S	OIS	5.10	4.03	3.95	**9.39**	**0.003**
	TIS	5.19	3.50	2.96		
BPRS	OIS	64.28	47.40	44.35	1.91	0.171
	TIS	66.38	41.33	35.81		
WHOQOL (physical)	OIS	57.35	63.05	66.08	1.10	0.298
	TIS	53.07	62.69	63.27		
WHOQOL (psychological)	OIS	53.28	62.33	66.58	**8.73**	**<0.05**
	TIS	46.15	57.15	58.52		
WHOQOL (social)	OIS	55.30	63.68	66.05	**4.45**	**<0.05**
	TIS	48.40	52.98	54.85		
WHOQOL (environmental)	OIS	56.35	65.35	67.13	0.65	0.422
	TIS	53.77	64.02	64.96		
BIS-11 (Attentional)	OIS	24.00	21.50	20.58	**8.88**	**<0.005**
	TIS	22.08	19.13	18.79		
BIS-11 (Motor)	OIS	29.23	24.88	23.33	1.59	0.211
	TIS	27.98	23.40	22.44		
BIS-11 (Planning)	OIS	33.85	29.03	26.83	**4.38**	**<0.05**
	TIS	30.75	26.15	25.31		
BIS-11 Total	OIS	87.05	75.55	70.73	**6.00**	**<0.05**
	TIS	80.79	68.69	66.58		
ACES	OIS	2.38	3.08	3.45	0.38	0.538
	TIS	2.02	3.15	3.56		
VAS frequency	OIS	7.79	3.26	4.89	**5.82**	**<0.05**
	TIS	7.15	2.97	2.79		
VAS intensity	OIS	7.32	3.13	4.89	**4.12**	**<0.05**
	TIS	7.12	2.91	2.85		

Significant results in **bold**. *Abbreviations:* ACES, Agitation–Calmness Evaluation Scale; BIS-11, Barratt Impulsiveness Scale; BL, baseline; BPRS, 242-item Brief Psychiatric Rating Scale; CGI-S, Clinical Global Impressions–Severity scale; GLM, general linear model; OIS, one-injection start; TIS, two-injection start; VAS, visual analogue scale; WHOQOL, World Health Organization Quality-of-Life Scale.

**Table 4 ijms-26-01394-t004:** Differences between patients on OIS (*n* = 13) and those assigned to TIS (*n* = 13) in their serum aripiprazole plus dehydroaripiprazole levels (means, in ng/mL); Student’s *t*-test.

	OIS	TIS	*t*	*p*
Aripiprazole baseline	*355.05*	340.13	1.12	0.301
Aripiprazole 1-week	*415.21*	287.68	3.38	0.078
Aripiprazole 2-weeks	*381.87*	249.23	5.71	**0.025**
Aripiprazole 4-weeks	302.95	220.89	2.71	0.114
Dehydroaripiprazole baseline	77.35	91.07	3.28	0.082
Dehydroaripiprazole 1-week	119.02	98.65	3.68	0.067
Dehydroaripiprazole 2-weeks	109.08	76.18	13.25	**<0.001**
Dehydroaripiprazole 4-weeks	89.66	75.88	0.99	0.331

*Abbreviations*. OIS, one-injection start; *p*, statistical significance probability; *t*, Student’s *t*-test; TIS, two-injection start. *Note:* Therapeutic window: aripiprazole, 100–350 ng/mL. Significant differences in **bold**. Supratherapeutic concentrations in *italics*.

**Table 5 ijms-26-01394-t005:** Differences between patients receiving aripiprazole LAI OIS regimen (*n* = 13) and those receiving aripiprazole LAI TIS regimen (*n* = 13) in their serum active moiety (aripiprazole plus dehydroaripiprazole) levels (in ng/mL); Student’s *t*-test.

	OIS	TIS	*T*	*p*
Baseline	432.40	431.20	3.29	0.082
1- week	*534.23*	386.34	4.64	**0.042**
2- week	490.95	325.41	6.14	**0.021**
4- week	392.61	273.94	2.50	0.127

*Abbreviations*. OIS, one-injection start; *p*, statistical significance probability; *t*, Student’s *t*-test; TIS, two-injection start. *Note:* Therapeutic window: Active moiety, 150–500 ng/mL. Significant differences in **bold**. Supratherapeutic concentrations in *italics*.

**Table 6 ijms-26-01394-t006:** Treatment associated with aripiprazole LAIs at discharge.

TIS (*n* = 86)	OIS (*n* = 86)
Antipsychotics	
Amisulpride	Quetiapine (3)
Haloperidol	Brexpiprazole
Clonazepam	Olanzapine
	Promazine
	Trifluoperazine
Antidepressants	
Trazodone (3)	Trazodone (3)
Vortioxetine	Duloxetine
Fluoxetine	Sertraline
	Amitriptyline
Anxiolytics/hypnotics	
Lorazepam (4)	Lorazepam (3)
Delorazepam (2)	Delorazepam (3)
Zolpidem (2)	Flurazepam (2)
	Diazepam
Channel blocking anti-anxiety agents	
Gabapentin	Gabapentin (2)
Pregabalin	
Mood stabilisers	
Lithium sulfate (2)	Lithium sulfate (4)
Lithium carbonate	Lithium carbonate (4)
Valproate	Valproate (4)
Lamotrigine (2)	Lamotrigine (2)
SUD-specific drugs	
Methadone	Naltrexone (3)
Drugs for other comorbidities	
Pantoprazole	

Note: Numbers in parentheses refer to the number of patients receiving that specific drug.

## Data Availability

Anonymised data will be made available upon reasonable request to the corresponding authors.

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
