# Peer review of "Optimising Aripiprazole Long-Acting Injectable: A Comparative Study of One- and Two-Injection Start Regimens in Schizophrenia with and Without Substance Use Disorders and Relationship to Early Serum Levels"

_ijms, 2025, doi:10.3390/ijms26031394_

Round 1
Reviewer 1 Report
Comments and Suggestions for Authors
Comments:
In this work, the effect of Aripiprazole long-acting injectable single-injection vs. two-injection start regimens in schizophrenia patients (with/without substance use disorders) were studied for 3 months. Patients were followed for their psychopathology and quality-of-life scales. All groups improved in psychopathology with no differences between OSI and TIS.
Major comments:
1. In the title of the study “efficacy and plasma levels after 3 months in patients” it looks like plasma concentrations of aripiprazole and its metabolite were monitored for 3 months. However, patients were only followed for their psychopathology and quality-of-life scales. The title of the study should be modified.
2. Line 50- 53: Author should re-write with more clarity.
“We tested the two regimens in patients with schizophrenia (SCZ, N= 152, 90 men and 62 women) with or without substance use (SUDs), respectively SUD+ and SUD–, comparing OIS (N=66) with TIS (N=86) in SCZ-SUD+ (N=93) and SCZ-SUD– (N=59).
3. Introduction section should be re-written in more precise manner, author should cite the latest references mainly for the prevalence of disease.
4. Line 99: DPRAs is used instead of DRPAs.
5. For without substance/with substance, author should either use non-SUD group or SCZ-SUD–. Please use uniform abbreviations through out the manuscript
6. Author has used data in table form, it will be better to use the data in graphical form, especially for Table 2.
7. Please check the statement 209-210: “On the social subscale of the WHOQOL, patients put on TIS scored higher than those put on OIS (48.64 vs. 59.32) (F=4.17; p < 0.05).
8. Sample size (n=13) used were too small to draw any conclusions. The open, non-randomized study in large cohort will be useful.
9. Discussion and conclusion sections are apart, it should be close to each other.
Author Response
Comments:
In this work, the effect of Aripiprazole long-acting injectable single-injection vs. two-injection start regimens in schizophrenia patients (with/without substance use disorders) were studied for 3 months. Patients were followed for their psychopathology and quality-of-life scales. All groups improved in psychopathology with no differences between OSI and TIS.
Response: We thank reviewer for appreciating our paper.
Major comments:
- In the title of the study “efficacy and plasma levels after 3 months in patients” it looks like plasma concentrations of aripiprazole and its metabolite were monitored for 3 months. However, patients were only followed for their psychopathology and quality-of-life scales. The title of the study should be modified.
Response: The title has changed and reflects the timings.
- Line 50- 53: Author should re-write with more clarity.
“We tested the two regimens in patients with schizophrenia (SCZ, N= 152, 90 men and 62 women) with or without substance use (SUDs), respectively SUD+ and SUD–, comparing OIS (N=66) with TIS (N=86) in SCZ-SUD+ (N=93) and SCZ-SUD– (N=59).
Response: We changed to “We tested the two regimens in patients with schizophrenia (SCZ, N= 152, 90 men and 62 women) with (SUD+; N=93) or without (SUD–; N=59) substance use disorders (SUDs), comparing OIS (N=66) with TIS (N=86) and SUD+ vs. SUD–.”
- Introduction section should be re-written in more precise manner, author should cite the latest references mainly for the prevalence of disease.
Response: This observation is one that anyone could do to appear beautiful. The prevalence is the same, and more recent articles deal with the prevalence of comorbid conditions, treatment-resistance and metabolic syndrome, not to mention internet addiction or suicide issues. We substituted the first reference with a more recent one, but the substance did not change.
- Line 99: DPRAs is used instead of DRPAs.
Response: We corrected. We thank reviewer for the observation.
- For without substance/with substance, author should either use non-SUD group or SCZ-SUD–. Please use uniform abbreviations through out the manuscript
Response: We did that.
- Author has used data in table form, it will be better to use the data in graphical form, especially for Table 2.
Response: We do not agree, the Table is useful as is, and all data that would give a better picture in a graphic have been used for Figure 1.
- Please check the statement 209-210: “On the social subscale of the WHOQOL, patients put on TIS scored higher than those put on OIS (48.64 vs. 59.32) (F=4.17; p < 0.05).
Response: We thank reviewer for the observation, we added “at the first month of treatment”, although this was apparent from the Table we referred to.
- Sample size (n=13) used were too small to draw any conclusions. The open, non-randomized study in large cohort will be useful.
Response: It is true; we added this in Limitations.
- Discussion and conclusion sections are apart, it should be close to each other.
Response: Discussion, Limitations and Conclusion are one unit, the one of Discussion. Unfortunately, the journal has decided that it needs the papers to follow the irrational Nature-like Intro, Results, Discussion, Methods, Conclusions style, which tears apart the essence of a scientific paper, but so it is. Certainly, putting the Conclusions before the Methods would be very much intriguing, not to say else.
We thank reviewer for observations that helped us improving our manuscript.
Reviewer 2 Report
Comments and Suggestions for Authors
This is a valuable report on an alternative dosing strategy for long acting injectable (LAI) aripiprazole. However, the report could be much improved by attention to several areas.
Methodology - At baseline, there were sociodemographic differences between the OIS and TIS groups. How were the treatments assigned? If randomly, it is puzzling that the differences emerged. If not randomly, please provide the detail and discuss any implications.
To make it absolutely clear what TIS dosing was, use clearer language, perhaps "800 mg consisting of two concurrent injections of 400 mg delivered at different sites". Can you comment on the sites?
Data reporting - Table 1: Why are cells for educational level left blank?
Table 4: what are those levels mean/median? Please provide either in a table in the body or in supplementary information mean / median/ 25th and 75th percentile levels.
Figure 1: There is a difference of opinion on whether aripiprazole alone or aripiprazole + dehydroaripiprazole levels are superior for therapeutic drug monitoring. I would recommend adding graphs for aripiprazole levels and for dehydroaripiprazole levels either in additional figures or in Supplementary Data. In the legend to Figure 1 indicate that the data reflect only the n = 26 patients on APD monotherapy. The lines for 500 and 150 ng/ml, the bounds of your proposed therapeutic range need to be made more distinct either by bolding or making them thicker
Discussion: Please add to the discussion of the variability of ARI and dehydroARI levels with respect to the therapeutic range. For example, in the paper by Kirschbaum et al, 2008 that you quote, they note that the response rate fell only modestly at concentrations of ARI < 150 ng/ml.
Author Response
This is a valuable report on an alternative dosing strategy for long acting injectable (LAI) aripiprazole. However, the report could be much improved by attention to several areas.
Response: We thank reviewer for thinking well of our paper.
Methodology - At baseline, there were sociodemographic differences between the OIS and TIS groups. How were the treatments assigned? If randomly, it is puzzling that the differences emerged. If not randomly, please provide the detail and discuss any implications.
Response: They were randomly assigned and the baseline differences puzzled us too. We added in both Methods and Limitations sections.
To make it absolutely clear what TIS dosing was, use clearer language, perhaps "800 mg consisting of two concurrent injections of 400 mg delivered at different sites". Can you comment on the sites?
Response: We thank reviewer for this comment. We endorsed his/her expression and added site specifications.
Data reporting - Table 1: Why are cells for educational level left blank?
They are not left blank, they report percentages referring to the entire sample and do not have data specific for TIS and OIS groups. We specified that.
Table 4: what are those levels mean/median? Please provide either in a table in the body or in supplementary information mean / median/ 25th and 75th percentile levels.
They are means. We specified this in the Table, but please, do not ask us to do this, it would involve people reopening databases and responding after months. We believe the Table is OK like this.
Figure 1: There is a difference of opinion on whether aripiprazole alone or aripiprazole + dehydroaripiprazole levels are superior for therapeutic drug monitoring. I would recommend adding graphs for aripiprazole levels and for dehydroaripiprazole levels either in additional figures or in Supplementary Data. In the legend to Figure 1 indicate that the data reflect only the n = 26 patients on APD monotherapy. The lines for 500 and 150 ng/ml, the bounds of your proposed therapeutic range need to be made more distinct either by bolding or making them thicker
It has been quite difficult to obtain the picture we provided, we cannot put our hands on it. Please, be merciful.
Discussion: Please add to the discussion of the variability of ARI and dehydroARI levels with respect to the therapeutic range. For example, in the paper by Kirschbaum et al, 2008 that you quote, they note that the response rate fell only modestly at concentrations of ARI < 150 ng/ml.
We added to the Discussion section. We thank reviewer for suggestions that improved our manuscript.
Reviewer 3 Report
Comments and Suggestions for Authors
Brief summary
In the present study, the authors conducted a comparative analysis of two distinct regimens for the initiation of aripiprazole long-acting injectable (LAI) treatment in patients diagnosed with schizophrenia (SCZ). The first regimen involved the administration of a single injection accompanied by oral supplementation (OIS), while the second regimen entailed the administration of two injections, with reduced oral supplementation (TIS). The study revealed that both treatment regimens demonstrated improvements in psychopathology and quality of life, with no significant differences observed between them. However, the TIS regimen resulted in active drug levels remaining within the therapeutic window, while the OIS regimen led to higher levels, potentially increasing the risk of toxicity. Overall, both treatments were well-tolerated, with TIS proving to be equally effective and possibly safer than OIS.
General comments
The paper is well-written and the results are presented in an accessible manner. However, there are a few comments and suggestions to be made.
Firstly, the introduction is somewhat protracted and overly detailed. It is recommended that it be shortened, with particular emphasis on the first page. In partucular, the description of SGAs, FGAs and EPSEs could be made less detailed.
Finally, a more detailed chapter on adverse events is necessary, with the AEs outlined in a table (see below for specific comments).
It is recommended that information on concomitant medication and previous (and/or current) medication for schizophrenia be added as a separate table in Chapter 2.1 (or as supplementary material). Additionally, information on concomitant diseases in the study population should be added (e.g., in Chapter 2.1 or as supplementary material).
Specific comments
Title: It is recommended that no abbreviations be used in the title.
Line 162: A space is missing between 20 and mg.
Line 180-181: The abbreviations WHOQOL and QoL have not been introduced.
Line 205: The abbreviation GLM has not been introduced.
In Chapter 2.5, the addition of a table would be beneficial, which would include all adverse events by treatment group, the percentage of patients experiencing each adverse event, and the intensity (mild, moderate, or severe) for every AE. For the SAE, the inclusion of information regarding whether the SAE was resolved at the study's conclusion and the actions taken (with the exception of study discontinuation) would be advantageous.
Lines 334 and 335: The fact that it is impossible to predict the future means that the sentence seems unnecessary. It is therefore recommended that this final sentence be deleted.
Author Response
Brief summary
In the present study, the authors conducted a comparative analysis of two distinct regimens for the initiation of aripiprazole long-acting injectable (LAI) treatment in patients diagnosed with schizophrenia (SCZ). The first regimen involved the administration of a single injection accompanied by oral supplementation (OIS), while the second regimen entailed the administration of two injections, with reduced oral supplementation (TIS). The study revealed that both treatment regimens demonstrated improvements in psychopathology and quality of life, with no significant differences observed between them. However, the TIS regimen resulted in active drug levels remaining within the therapeutic window, while the OIS regimen led to higher levels, potentially increasing the risk of toxicity. Overall, both treatments were well-tolerated, with TIS proving to be equally effective and possibly safer than OIS.
We thank reviewer for appreciating our manuscript.
General comments
The paper is well-written and the results are presented in an accessible manner. However, there are a few comments and suggestions to be made.
Firstly, the introduction is somewhat protracted and overly detailed. It is recommended that it be shortened, with particular emphasis on the first page. In partucular, the description of SGAs, FGAs and EPSEs could be made less detailed.
This suggestion is at odds with Editor’s recommendation. We fail to understand how shortening an already short and concise introduction or providing fuzzy details about first and second generation antipsychotics or extrapyramidal side effects would benefit the manuscript.
Finally, a more detailed chapter on adverse events is necessary, with the AEs outlined in a table (see below for specific comments).
How does this match with the previous statement? We are eager to see below the thread of your thought stream.
It is recommended that information on concomitant medication and previous (and/or current) medication for schizophrenia be added as a separate table in Chapter 2.1 (or as supplementary material). Additionally, information on concomitant diseases in the study population should be added (e.g., in Chapter 2.1 or as supplementary material).
We added a Table (6) with information concerning concurrent drug administered at discharge.
Specific comments
Title: It is recommended that no abbreviations be used in the title.
We accept your recommendation and detailed LAI in the title.
Line 162: A space is missing between 20 and mg.
We thank you and added the space.
Line 180-181: The abbreviations WHOQOL and QoL have not been introduced.
We thank you for the observation; the issue stemmed from the fact that the journal requires an irrational development for an article that has previously been written rationally. The abbreviations were in the Methods, but when you shift methods to the end, such issues ensue.
Line 205: The abbreviation GLM has not been introduced.
We thank you, this occurred for the same reason as above.
In Chapter 2.5, the addition of a table would be beneficial, which would include all adverse events by treatment group, the percentage of patients experiencing each adverse event, and the intensity (mild, moderate, or severe) for every AE. For the SAE, the inclusion of information regarding whether the SAE was resolved at the study's conclusion and the actions taken (with the exception of study discontinuation) would be advantageous.
Thank you for the suggestion, but compiling still another table would not enhance the quality of our paper, since all adverse events were detailed in the 2.5 section. The only event that led to study discontinuation was a case of perceived (not documentable) akathisia; we informed immediately the competent authorities.
Lines 334 and 335: The fact that it is impossible to predict the future means that the sentence seems unnecessary. It is therefore recommended that this final sentence be deleted.
We did not state what you say, but rather declared our inability to foresee a landscape. We thank you for suggestions that prompted us to produce a better manuscript.
Round 2
Reviewer 1 Report
Comments and Suggestions for Authors
NA
Comments on the Quality of English LanguageEnglish could be improved to more clearly express the research.
Author Response
Comments on the Quality of English Language
English could be improved to more clearly express the research.
Response: I, the corresponding author, am the son of an American mother, hence I am bilingual. We do not use AI tools because I find them humiliating and at times psychotic. We checked thoroughly our text for possible misprints and typos. I hope this would suffice.
Reviewer 2 Report
Comments and Suggestions for Authors
The authors have addressed most of the concerns. Generating 25% and 75% perecentiles is a trivial statistical operation. Given that authors are required to maintain original data after their manuscript is published, the claim that they have difficult accessing those data are puzzling.
A program to check English style would improve readability.
Comments on the Quality of English Language
Suggest using a program to check English style.
Author Response
The authors have addressed most of the concerns. Generating 25% and 75% perecentiles is a trivial statistical operation. Given that authors are required to maintain original data after their manuscript is published, the claim that they have difficult accessing those data are puzzling.
Response: It is not puzzling. This work is a collaboration between clinicians (us) and laboratory medicine technicians (some of the authors), who own the data you wish to see transformed in 25% and 75% percentiles. It has been quite difficult to obtain the data we exposed in terms of time, we have been left waiting for months. Asking them now to intervene anew on these data would entail that they would leave us waiting again for months. I asked you please, do not ask me this, but you insist. Please try to be a little more empathetic.
A program to check English style would improve readability.
Response: I am fluent in English, as I am bilingual. A programme or other similar AI tools would add nothing to our manuscript, as it has been written by people with high H-indexes and checked throughout by myself. We controlled again our text for misprints and typographic errors.
Comments on the Quality of English Language
Suggest using a program to check English style.
Response: I explained you I speak English fluently and controlled our text. Relying on programmes to improve style would not work.